# Responses to Medical Treatment in 192 Patients with Pancreatic Neuroendocrine Neoplasms Referred to the Copenhagen Neuroendocrine Tumour Centre in 2000–2020

**DOI:** 10.3390/cancers16061190

**Published:** 2024-03-18

**Authors:** Sofie Skovlund Petersen, Stine Møller, Cecilie Slott, Jesper Krogh, Carsten Palnæs Hansen, Andreas Kjaer, Pernille Holmager, Peter Oturai, Rajendra Singh Garbyal, Seppo W. Langer, Ulrich Knigge, Mikkel Andreassen

**Affiliations:** 1ENETS Center of Excellence, Copenhagen University Hospital—Rigshospitalet, 2100 København, Denmark; sofie.skovlund@live.dk (S.S.P.); stine.moeller.02@regionh.dk (S.M.); cecilie.slott@regionh.dk (C.S.); jesper.krogh@dadlnet.dk (J.K.); carsten.palnaes.hansen@regionh.dk (C.P.H.); akjaer@sund.ku.dk (A.K.); pernille.holmager.01@regionh.dk (P.H.); peter.sandor.oturai@regionh.dk (P.O.); rajendra.singh.garbyal@regionh.dk (R.S.G.); seppo.langer@regionh.dk (S.W.L.); ulrich.peter.knigge@regionh.dk (U.K.); 2Department of Endocrinology and Metabolism, Copenhagen University Hospital—Rigshospitalet, 2100 København, Denmark; 3Department of Clinical Medicine, University of Copenhagen, 1172 København, Denmark; 4Department of Surgery and Transplantation, Copenhagen University Hospital—Rigshospitalet, 2100 København, Denmark; 5Department of Clinical Physiology and Nuclear Medicine & Cluster for Molecular Imaging, Copenhagen University Hospital—Rigshospitalet, 2100 København, Denmark; 6Department of Biomedical Sciences, University of Copenhagen, 1172 København, Denmark; 7Department of Pathology, Copenhagen University Hospital—Rigshospitalet, 2100 København, Denmark; 8Department of Oncology, Copenhagen University Hospital—Rigshospitalet, 2100 København, Denmark

**Keywords:** pancreatic neuroendocrine tumors, treatment efficacy, somatostatin analogue, peptide receptor radionuclide therapy, everolimus, chemotherapy

## Abstract

**Simple Summary:**

Pancreatic neuroendocrine tumors are a rare and heterogenous group of neoplasms. Surgical resection is the only curative option. However, there has been an increase in palliative medical options. The aim of this retrospective study was to investigate responses for the most commonly used medical treatments in 192 patients. The current results support the effectiveness of somatostatin analogues in low-grade tumors and showed that it might also be used in patients with Ki-67 ≥ 10%. Treatment with streptozocin and 5-floururacil as first-line treatment showed good efficacy for G2 disease. Due to good efficacy and generally good tolerability PRRT might be considered as first-line treatment for NET G2. The results confirmed poor prognosis in high-grade tumors treated with carboplatin/etoposide or temozolomide. The current results provide valuable knowledge as current treatment algorithms and sequencing are primarily guided by expert opinions with limited evidence.

**Abstract:**

Background: Given the rarity and heterogeneity of pancreatic neuroendocrine neoplasms (pNEN), treatment algorithms and sequencing are primarily guided by expert opinions with limited evidence. Aim: To investigate overall survival (OS), median progression-free survival (mPFS), and prognostic factors associated with the most common medical treatments for pNEN. Methods: Retrospective single-center study encompassing patients diagnosed and monitored between 2000 and 2020 (n = 192). Results: Median OS was 36 (95% CI: 26–46) months (99 months for grade (G) 1, 62 for G2, 14 for G3, and 10 for neuroendocrine carcinomas). Patients treated with somatostatin analogues (SSA) (n = 59, median Ki-67 9%) had an mPFS of 28 months. Treatment line (HR (first line as reference) 4.1, 95% CI: 1.9–9.1, *p* ≤ 0.001) emerged as an independent risk factor for time to progression. Patients with a Ki-67 index ≥10% (n = 28) had an mPFS of 27 months. Patients treated with streptozocin/5-fluorouracil (STZ/5FU) (n = 70, first-line treatment n = 68, median Ki-67 10%) had an mPFS of 20 months, with WHO grade serving as an independent risk factor (HR (G1 (n = 8) vs. G2 (n = 57)) 2.8, 95% CI: 1.1–7.2, *p*-value = 0.031). Median PFS was 21 months for peptide receptor radionuclide therapy (PRRT) (n = 41, first line n = 2, second line n = 29, median Ki-67 8%), 5 months for carboplatin and etoposide (n = 66, first-line treatment n = 60, median Ki-67 80%), and 3 months for temozolomide-based therapy (n = 56, first-line treatment n = 17, median Ki-67 30%). Conclusion: (1) Overall survival was, as expected, highly dependent on grade; (2) median PFS for SSA was around 2.5 years without difference between tumors with Ki-67 above or below 10%; (3) STZ/5FU as first-line treatment exhibited a superior mPFS of 20 months compared to what has historically been reported for targeted treatments; (4) PRRT in G2 pNEN achieved an mPFS similar to first-line chemotherapy; and (5) limited treatment efficacy was observed in high-grade tumors when treated with carboplatin and etoposide or temozolomide.

## 1. Introduction

Pancreatic neuroendocrine neoplasms (pNEN) represent a rare and heterogeneous group of neoplastic disorders [1,2]. Clinical presentations of pNEN include slow-growing asymptomatic incidental findings, symptomatic functioning tumors secreting various hormones, and highly aggressive tumors with very poor prognosis [3,4]. Pancreatic NEN are classified based on their degree of differentiation in well-differentiated neuroendocrine tumors (NET) and poorly differentiated neuroendocrine carcinomas (NEC). In addition, NET are graded based on the mitotic rate and Ki67 index as NET grade (G) 1 (Ki-67 < 3%), NET G2 (Ki-67 = 3–20%), and NETG3 (Ki-67 > 20%), whereas NEC are subdivided into large- and small-cell NEC based on morphological features [5].

Surgical resection remains the only curative option for pNEN. However, many patients are diagnosed in an advanced stage and, therefore, there has been a growing array of treatment options for patients with unresectable or metastatic pNEN [6]. The choice of treatment depends on the tumor grade and clinical stage of the patient [6,7,8]. However, due to the rarity and heterogeneity of pNEN, treatment algorithms, including the types and sequencing of different modalities, predominantly rely on expert consensus and are supported by limited empirical evidence. Local practices also appear to influence treatment decisions, as exemplified by the use of peptide receptor radionuclide therapy (PRRT), which has been employed in Europe for over two decades but has only recently gained approval in the United States [9].

This study presents data from 192 pNEN patients encompassing the entire spectrum of the disease, who received medical treatment for unresectable disease or residual tumor after resection with a palliative or curative intend. The primary objectives were to determine median overall survival (OS) and the median progression-free survival (mPFS) including prognostic factors associated with the most common medical treatments.

## 2. Materials and Methods

### 2.1. Study Population

The study cohort comprised consecutive patients diagnosed with pNEN who received medical treatment at Rigshospitalet from 1 January 2000 to 30 December 2020. Tumor grade was determined from the Ki-67 index. The diagnosis of NEC was established through cyto-histomorphology, Ki-67 proliferation index, immunohistochemistry, and supported by mutational analysis. All patients underwent evaluation for surgery with either a curative or palliative intent. Medical treatment was only considered if the patient’s disease was deemed unresectable at diagnosis or some point in time during follow-up or if the patient was deemed inoperable due to factors such as overall health condition.

### 2.2. Data Acquisition

At the time of diagnosis, baseline data were prospectively collected and recorded in a dedicated database. This information encompassed demographic details such as age and gender, clinical presentation (including incidentaloma or symptomatic tumor, hormone secretion, and germline mutations), tumor characteristics (size and location within the pancreas), immunohistochemistry (Ki-67, CgA, synaptophysin, and somatostatin receptor type 2 (UMB1)), and somatic mutations obtained via next-generation sequencing (NGS). The stage of disease at baseline was classified based on pathological findings from either biopsy or surgery as well as from imaging. Based on these data, patients were categorized into three groups: (1) local disease, (2) regional disease with lymph node metastases, and (3) disseminated disease with distant metastases.

As part of the study protocol, follow-up data, including treatment modalities and PFS for all medical treatments, were documented through the electronic medical record system. No patients were lost to follow-up, due to the unique national identification number assigned to every citizen and resident in Denmark and the nationwide electronic medical record system, which provided unrestricted access to patients’ records. All patients were monitored until death or the conclusion of follow-up on 31 December 2021.

### 2.3. Outcome

Primary outcomes were median OS for the entire cohort and mPFS and prognostic factors specific to the various treatments. The mPFS was determined from the initiation of treatment to the point of progression. Progression was defined at death of any course, clinical progression, or radiological progression assessed through routine clinical practice. Medical treatments used exclusively for purposes other than disease control, such as somatostatin analogues employed to reduce hormonal excess, were excluded from the statistical analyses. Overall disease-specific survival and median recurrency-free survival after intended radical surgery for this cohort have previously been published [10].

### 2.4. Statistics

Descriptive statistics were employed to summarize demographic and clinical characteristics. Categorical variables were presented as frequencies and percentages. Continuous variables are expressed as mean and standard deviation (SD) or as median and quartiles, when appropriate. To achieve an approximate normal distribution, p-CgA and Ki-67 index were Log2 transformed.

To explore potential associations with mPFS, univariable Cox regression analyses were conducted for the following variables: gender, age at diagnosis, year of diagnosis (2000–2009 vs. 2010–2020), tumor grade, functional status of the tumor, pancreatic location (head vs. tail), Log2(CgA), Log2(Ki-67), disease stage, size of the primary tumor, primary surgical intervention, and treatment line (first line vs. subsequent lines). All variables with a *p*-value ≤ 0.2 were included in multivariable Cox regression analyses (backward elimination (conditional)). Age and gender were included in all multivariable Cox regression analyses. Each multivariable Cox regression analysis was performed twice, first with proliferation expressed as the Ki-67 index and subsequently with proliferation based on WHO grading instead of the Ki-67 index. All results from the analyses are reported as hazard ratios (HR) with 95% confidence intervals and corresponding *p*-values. To prevent overfitting of the statistical models, Cox regression analyses were exclusively conducted for treatments administered to 40 or more patients. Furthermore, PFS was estimated by Kaplan–Meier curves, and group differences were assessed using the Log rank test. *p*-values ≤ 0.05 were considered statistically significant. All statistical analyses were carried out using IBM SPSS Statistics software (version: 28.0.0.0).

## 3. Results

### 3.1. Baseline Characteristics

The cohort included 192 patients (for details, see Table 1). A total of 45 patients (23%) had surgery with a curative intent but developed at some point in time unresectable metastatic disease; median recurrence-free survival was 26 (range 4–58) months. Seven (4%) had palliative surgery at diagnosis or during follow-up.

Median follow-up period was 28 (10–61) months. The mean age at time of diagnosis was 64 ± 13 years. Four patients had a mutation in the Menin gene and one was diagnosed with Von Hippel–Lindau disease. Eight patients had a functioning tumor. Based on the WHO classification of NEN, 17/187 (9%) were categorized as NET G1, 90/187 (48%) as NET G2, 20/187 (11%) as NET G3, and 60/187 (32%) as NEC. The median Ki-67 proliferation index was 16 (8–50)%. In five patients, Ki-67 index was not available. The median plasma CgA was 322 (98–1020) pmol/L. Localized disease was present in 26 (14%) patients, regional disease with lymph node metastases in 22 (11%), and disseminated disease with distant metastases in 144 (75%).

### 3.2. Overall Survival (OS)

Overall survival and OS stratified by WHO grade are shown in Figure 1A and Figure 1B, respectively. Median OS was 36 (95% CI: 26–46) months, 99 (95% CI: 65–133) months for patients with NET G1, 62 (95% CI: 44–80) months for patients with NET G2, 14 (95% CI: 0–28) months for patients with NET G3, and 10 (95% CI: 8–12) months for patients with NEC.

### 3.3. Somatostatin Analogues

A total of 59 patients (31%) received treatment with somatostatin analogues (SSA). Among these, 56 were treated with lanreotide autogel (minimum dose of 120 mg every 4 weeks) and 3 with octreotide acetate (minimum dose of 30 mg every 4 weeks). A total of 37 out of the 59 patients (63%) received SSA as first-line treatment; 81% of the patients had NET G2 with a median Ki-67 index of 9 (5–12)%.

Progression-free survival and PFS stratified by WHO grade are illustrated in Figure 2A and Figure 2B, respectively. The mPFS was 28 (95% CI: 25–31) months. Patients with a Ki-67 index <10% (n = 30, median Ki-67 index 5 (2–8)%) had an mPFS of 29 (95% CI: 15–44) months, compared to 27 (95% CI: 24–30) months in patients with a Ki-67 index of 10% or higher (n = 28, median Ki-67 index 12 (10–15)%—HR (Ki-67 index < 10% reference) 1.4, 95% CI: 0.7–2.9, *p* = 0.32, Figure 3B). Potential risk factors for disease progression, identified through univariable Cox regression analyses, are detailed in Table 2. In multivariable Cox regression analyses, the line of treatment emerged as the only independent risk factor, with a beneficial impact of using SSA as first-line treatment (Table 2 and Figure 3A).

### 3.4. Streptozocin and 5-Fluorouracil

A total of 70 patients (36%) received treatment with a combination of streptozocin and 5-fluorouracil (STZ/5FU), in 61 as first-line therapy. The induction dose of STZ was 500 mg/m^2^ on day 1–5, combined with 5FU 400 mg/m^2^ on day 1–3 the first week, followed by STZ 1000 mg/m^2^ and 5FU at 400 mg/m² every third week. Most patients (81%) had NET G2 and median Ki-67 index 10 (5–15)%. The mPFS was 20 (95%-CI: 15–25) months (Figure 2C,D). Proliferation, expressed both by Ki-67 index and by grade, was identified as the only independent risk factor: HR (per 2-fold increase in Ki-67 index) 1.3, 95%-CI: 1.1–1.6, *p* = 0.008, HR (NET G1 vs. NET G2) 2.8, 95%-CI: 1.1–7.2, *p* = 0.031, and HR (NET G1 vs. NEC) 11.1, 95%-CI: 2.0–60.9, *p* = 0.005 (Table 3).

### 3.5. Peptide Receptor Radionuclide Therapy

A total of 41 patients (21%) were treated, with PRRT with only 2 receiving it as first-line treatment. A total of 29 patients received PRRT as second-line treatment, 9 as third-line treatment, and 1 as fourth-line treatment. A standard treatment regimen comprising four cycles of 7.4 GBq Lu-177-DOTATATE with an eight-week interval was used. For renal protection, an amino acid solution was co-administered following international guidelines [11]. Most patients (74%) had NET G2 and median Ki-67 index 8 (4–15)%. The mPFS was 21 (95%-CI: 15.5–26.5) months (Figure 2E,F). Possible risk factors for progression identified by univariable Cox regression analyses are presented in Table 4. When proliferation was expressed by Ki-67 index CgA (HR (per 2-fold increase in plasma CgA) 1.2, 95%-CI: 1.0–1.4, *p*-value = 0.011) and Ki-67 index (HR (per 2-fold increase in Ki-67 index) 1.4, 95%-CI: 1.0–1.8, *p*-value = 0.052), both were identified as independent risk factors for progression, including death. When the Ki-67 index was replaced with WHO grade, only CgA remained as an independent risk factor (HR (per 2-fold increase in plasma CgA) 1.2, 95%-CI: 1.0–1.4, *p* = 0.055).

### 3.6. Everolimus

A total of 24 patients (13%) were treated with everolimus but only 1 patient received this as first-line treatment. The target dosage was 10 mg per day. Most patients (79%) had NET G2 and median Ki-67 index 12 (7–18)%. Median PFS was 5 (95%-CI: 3–7) months (Figure 4E). As less than 40 patients received everolimus, neither univariable nor multivariable Cox regression analyses were conducted.

### 3.7. Temozolomide

A total of 56 patients (29%) were treated with temozolomide, in 46 patients as monotherapy and in 10 in combination with capecitabine (TemCap). In monotherapy, patients received temozolomide 200 mg/m^2^ daily on days 1–5 of a 28-day cycle. Temcap was administrated as capecitabine 750 mg/m^2^ orally twice daily on days 1–14 and temozolomide 150 mg/m^2^ divided into two doses daily on days 10–14 of a 28-day cycle. In total, 27 (48%) tumors were classified as NEC, 15 (27%) as NET G3, and 14 (25%) as NET G2; median Ki-67 index 30 (18–60)%. A total of 17 of the 56 patients (30%) received temozolomide as their first-line treatment, including 10 of the 15 patients with NET G3. The mPFS across all patients treated with temozolomide was 3 (95% CI: 3–3 months) (Figure 4C,D). The mPFS was 6 (95% CI: 4.2–7.8) months in NET G3 patients who received temozolomide as first-line treatment. The univariable Cox regression analysis did not identify any variables as potential risk factors for disease progression in this group of patients (Table 5A).

### 3.8. Other Treatments

Medical treatments that were administered to 20 or less patients included interferon alfa-2b (n = 9), sunitinib (n = 8), topotecan (n = 13), and capecitabine (n = 5). Data on median PFS and patient characteristics are presented in Appendix A.

### 3.9. Carboplatin and Etoposide

A total of 67 (35%) patients received treatment with a combination of carboplatin and etoposide, 60 as first-line treatment. The majority of patients (79%) had NEC. Patients received up to six cycles of IV carboplatin (AUC 5) day 1 and oral etoposide 200 mg/m^2^ divided into two doses days 1–3 of a 21-day cycle. The median Ki-67 index was 80 (40–90)%. Median PFS was 5 (95%-CI: 4–6) months (Figure 4A,B). The multivariable Cox regression analysis did not reveal any independent risk factors for progression (Table 5B).

### 3.10. Post Hoc Analyses

For patients with NET G2, the most common therapy sequence was STZ/5FU as first-line treatment, followed by four cycles of PRRT as either second (n = 21) or third (n = 5) line treatment. This led us to assess the combined efficacy of these two treatments (data available for 26 patients) and their potential impact on kidney function (data available for 24 patients), given that both are known to be nephrotoxic. Individual data on combined PFS are depicted in Figure 5. In six patients, two additional cycles of PRRT treatments were administered.

Renal function was assessed by a radiotracer plasma-clearance routine method used at our institution [12]. Mean renal clearance at baseline (prior to STZ/5FU) was 83 mL/min/1.73 m^2^, which decreased to 74 mL/min/1.73 m^2^ before the first PRRT cycle (*p* = 0.002, with a median interval of 18 (6–35) months between measurements) and further declined to 66 mL/min/1.73 m^2^ one year after the fourth PRRT cycle (*p* = 0.002, over an 18-month interval). These figures correspond to an annual mean reduction in kidney function of 10.2 mL/min/1.73 m^2^ (11.4%) during chemotherapy and 5.0 mL/min/1.73 m^2^ (6.9%) during PRRT (from the first PRRT cycle to one year post last PRRT cycle). The combined annual impact of both treatment modalities on renal function was a decrease of 6.3 mL/min/1.73 m^2^ (7.8%).

## 4. Discussion

Our main findings include (1) overall survival was, as expected, highly dependent on grade and (2) confirmation of mPFS of approximately 2.5 years for SSA in patients with low-grade tumors. Notably, SSA also appeared effective as first-line treatment in patients with a Ki-67 index ≥ 10%. (3) Administration of STZ/5FU as first-line treatment in patients with NET G2 demonstrated a PFS of 20 months, outperforming historical results achieved by targeted therapies. (4) The use of PRRT as second- or third-line treatment following chemotherapy in NET G2 cases showed promising results, with an mPFS of 21 months and (5) our study reaffirmed that patients with high-grade tumors treated with temozolomide-based chemotherapy or the combination of carboplatin and etoposide had short mPFS of around 6 months.

Lanreotide’s antiproliferative effect in pNET was established in the CLARINET study published in 2014 [13,14]. Both European and American guidelines now recommend SSA as first-line treatment for pNET G1 with stable disease or slow growth and G2 with low proliferation index [6,8]. Given the similarities in somatostatin receptor affinity, it is hypothesized that the antitumor activity observed is a class effect inherent to first-generation SSA [13]. The CLARINET study, which exclusively included patients with a Ki-67 index < 10%, reported an mPFS of 30 months in the pNET subgroup, aligning with the mPFS observed in our study [15]. Our findings further suggest that SSA could be a viable treatment option for NET G2 patients with a Ki-67 index ≥ 10%. Among the 59 patients treated with SSA, 28 had Ki-67 index ≥ 10% and exhibited mPFS comparable to those with a median Ki-67 < 10%. Only one study has explored the antiproliferative efficacy of SSA in pNET with a Ki-67 index ≥ 10%, finding an mPFS of 12 months, which was substantially shorter than the mPFS of 27 months observed in the present study [16]. In the multivariate analyses, the treatment line emerged as the only independent risk factor, indicating a favorable impact of SSA as a first-line treatment, while proliferation levels did not predict progression during treatment.

STZ/5FU is a well-established treatment and has been used for decades in pNEN [17,18]; yet, guidelines provide no clear recommendation for the sequencing of STZ/5FU compared to other treatment options. In European guidelines, STZ/5FU is recommended as an option for first-line treatment for NET G2 with high proliferation index and for NET G3 [6,8]. The North American Neuroendocrine Tumor Society (NANETS) recommends STZ/5FU as a treatment option for both NET G1 and G2 but does not specify its sequencing in comparison to other treatments [7]. In our study, patients with a median KI-67 index of 10% treated with STZ/5FU had an mPFS of 20 months, the majority of patients receiving STZ/5FU as first-line treatment. It is generally believed that the response rate to chemotherapy is higher in tumors with higher proliferation rates and this phenomenon seems also to hold true for pNET [19,20]. However, our study found shorter mPFS in tumors with increasing proliferation (HR for progression 1.3 per two-fold increase in Ki-67 index). This aligns with a retrospective study of 96 patients, which reported longer PFS in pNET with Ki-67 below 15% [21]. Nevertheless, these retrospective, nonrandomized studies do not definitively conclude whether the PFS reflect a generally better prognosis in low proliferative tumors or a better response to chemotherapy. The efficacy of STZ/5FU has never been investigated in a randomized controlled trial (RCT). Previous retrospective studies in patients with pNEN have reported similar results to ours [21,22]. In comparison, data from targeted treatments have shown a substantially lower mPFS, e.g., in RCT, the mPFS for both everolimus and sunitinib was around 11 months [23,24]. In our clinic, everolimus is usually reserved for third- or fourth-line treatment in pNET G2, which might explain the notably low mPFS of five months.

The use of PRRT in pNEN originates primarily from retrospective studies and smaller prospective phase II trials [6,7,8,9,25]. Results from the first RCT employing PRRT in pNET have recently been published and showed an mPFS for PRRT of 21 months vs. 11 months for sunitinib [26]. In our cohort, PRRT was mainly used as second-line treatment and, here, we found an mPFS of 21 months, which was one month longer than mPFS for STZ/5FU used as first-line treatment and in accordance with previous published data [27]. The retrospective NETTER-R trial (n = 110) published in 2022 found an mPFS of 25 months and reported a significantly longer mPFS and OS in patients who did not receive chemotherapy prior to PRRT [27]. In NANETS guidelines, PRRT is an option for G1 and G2 tumors but they do not provide guidance on treatment sequencing [7]. In the European Society for Medical Oncology (ESMO) and ENETS guidelines, PRRT is recommended as a treatment option for G1, G2, and NET G3 but not as first-line treatment [8,28,29]. The new ENETS guideline from 2023 recommends PRRT as second-line treatment (after SSA) in asymptomatic patients with slow-growing tumors [6].

Patients who received STZ/5FU as first-line therapy followed by PRRT had an annual decline in kidney function of 6.3 mL/min/1.73 m^2^. This rate of decline is approximately five times greater than the expected annual physiological decrease [30] despite the implementation of preventive measures such as adequate hydration and amino acid infusions prior to and simultaneously with PRRT [17]. During four cycles of PRRT and subsequent one year of follow-up, we observed an annual decrease in kidney function of 6.9%, which exceeds the previously reported rate for PRRT with ^177^Lu (annual loss of 3.8%) [31] and may be a consequence of prior treatment with STZ/5FU.

Recent advancements in the past decade have led to significant changes in treatment guidelines for high-grade NEN, particularly distinguishing between NET G3 and NEC [32]. The combination of temozolomide and capecitabine is now recommended as first-line treatment for NET G3 [6,7,8]. Overall, temozolomide-based treatment was associated with an mPFS of three months. As expected, subgroup analysis showed improved PFS in first-line setting and in NET G3 compared to later lines and NEC [33,34]. Consistent with European and American guidelines, the majority of the NEC patients received carboplatin and etoposide as first-line treatment [6,7,8]. The observed mPFS of five months is consistent with other studies, and we did not identify any prognostic factors that could predict treatment outcomes [28].

The strengths of this study are the large patient cohort and the comprehensive nature of the data, derived directly from patient files with no loss to follow-up. However, there are also some important limitations. Most importantly, the retrospective study design inherently faces risk of bias, such as selection bias. We attempted to mitigate this by employing multivariable analyses, but risk of residual confounding remains a concern. For example, we cannot exclude the possibility that the identification of treatment line (favorable impact of first-line treatment) as an independent risk factor for PFS of SSA is caused by factors not accounted for in the study. Furthermore, the retrospective design introduces the potential for missing data and changes in, e.g., radiological stage classification over time. There have also been changes in treatment guidelines during the study period, which have led to alterations in treatment sequencing and the introduction of new treatment options. Classification of patients was based on data from time of diagnosis. For some patients, disease progression has necessitated a reclassification of tumor grade, potentially leading to discrepancies in the accuracy of the WHO grade at the time of subsequent treatment lines. Since most patients underwent more than one type of treatment, there is an inevitable overlap of patients across different treatment groups. Finally, despite having up to 20 years of follow-up, a considerable proportion of our patients were diagnosed more recently and, thus, have short follow-up.

## 5. Conclusions

In conclusion, our study further supports the effectiveness of SSAs in managing low-grade tumors, demonstrating an mPFS of approximately 2.5 years. Importantly, and as a novel observation, our data showed that SSA might also be used in patients with Ki-67 ≥ 10%. We observed notable efficacy of STZ/5FU as first-line treatment for high-grade NET G2 disease, achieving a longer mPFS compared to targeted therapies. PRRT, when used as a second-line treatment following chemotherapy, showed promising outcomes, with an mPFS that was one month longer than that achieved with STZ/5FU as first-line therapy. Considering generally better tolerability, PRRT might be used as first-line treatment for NET G2. Finally, our study confirmed very short mPFS for patients with high-grade tumors treated with temozolomide-based therapy or combination of carboplatin and etoposide.

## Figures and Tables

**Figure 1 cancers-16-01190-f001:**
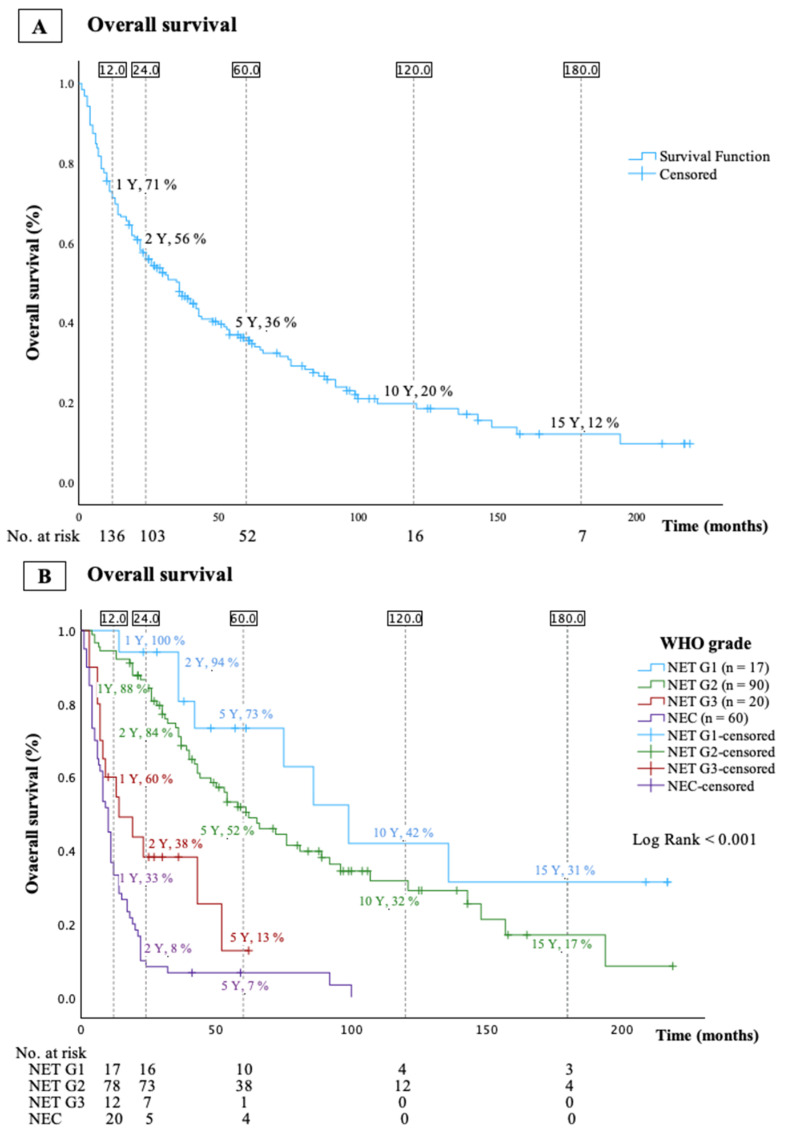
Kaplan–Meier curves presenting overall survival for the entire cohort (n = 192, (**A**)) and overall survival stratified by WHO grade (n = 187, (**B**)).

**Figure 2 cancers-16-01190-f002:**
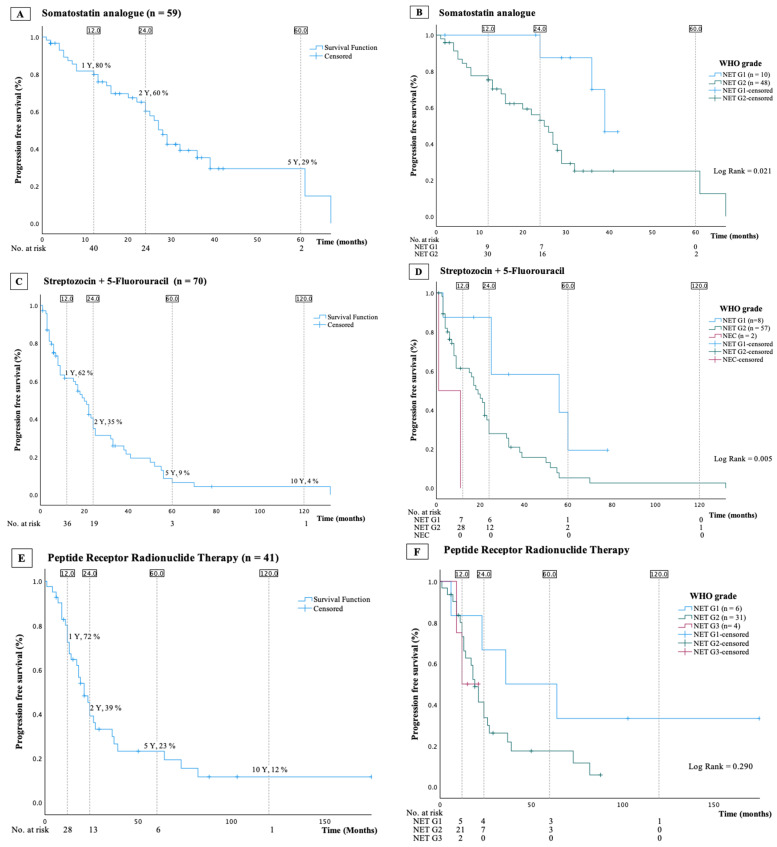
Kaplan–Meier curves presenting overall progression-free survival and progression-free survival stratified by WHO grade in patients treated with somatostatin analogue (**A**,**B**), with streptozocin/5-fluorouracil (**C**,**D**), or with peptide receptor radionuclide (**E**,**F**).

**Figure 3 cancers-16-01190-f003:**
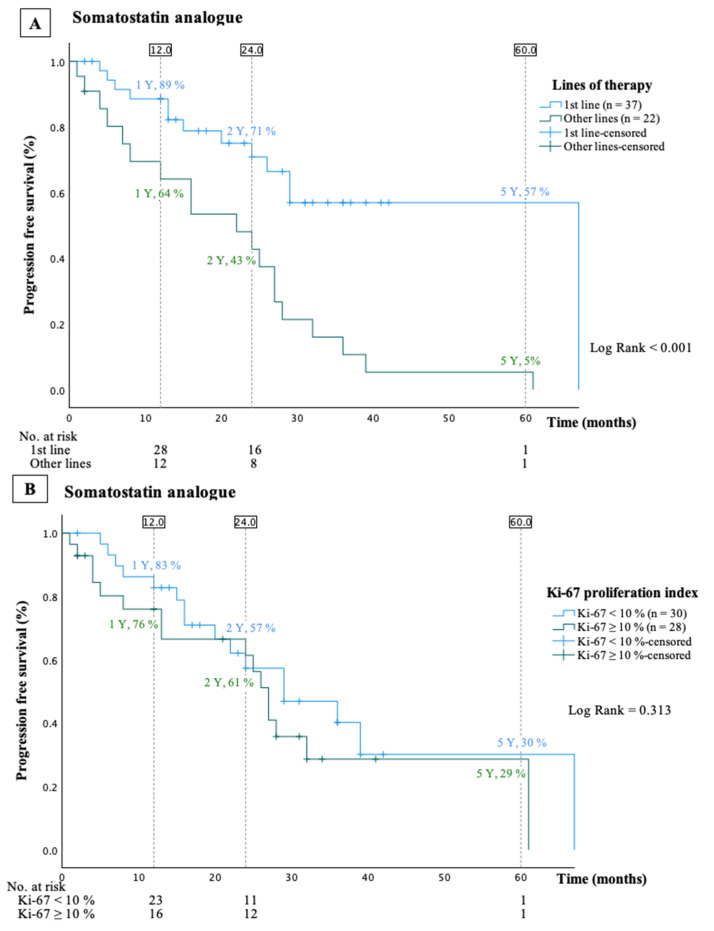
Kaplan–Meier curves presenting progression-free survival stratified by treatment line (**A**) or by Ki-67 index (**B**) in patients treated with somatostatin analogue.

**Figure 4 cancers-16-01190-f004:**
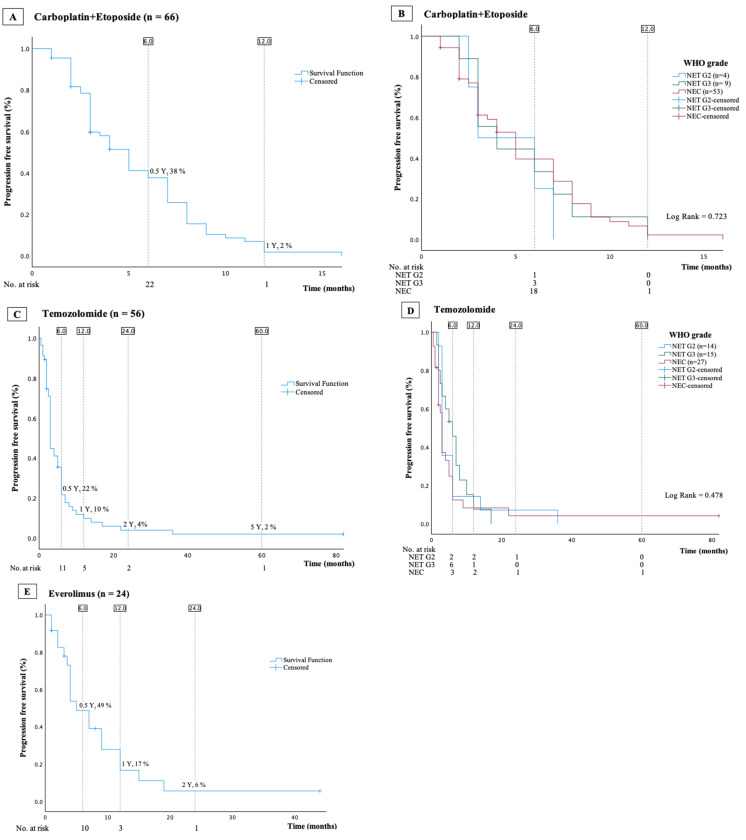
Kaplan–Meier curves presenting overall progression-free survival and progression-free survival stratified by WHO grade in patients treated with carboplatin and etoposide (**A**,**B**), with temozolomide (**C**,**D**), or with everolimus (**E**).

**Figure 5 cancers-16-01190-f005:**
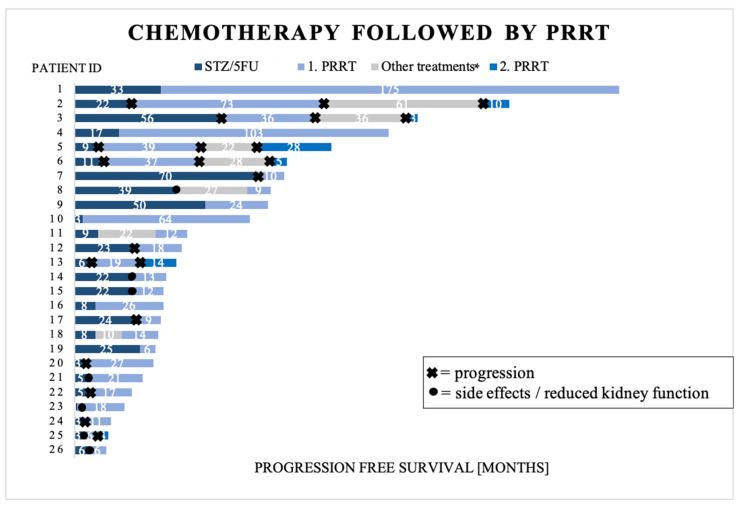
Therapy sequencing in 26 patients sorted by total progression-free survival in months. STZ/5FU = streptozocin/5-fluorouracil; PRRT = peptide receptor radionuclide therapy. First PRRT includes four treatment cycles. Second PRRT includes an additional two treatment cycles. * Patient number 2, 3, 5, 6, 8, and 11 received SSA as other treatment. Patient number 18 received interferon alfa-2b as 2nd-line treatment. Patient number 17 received carboplatin + etoposide as 2nd-line treatment (unknown efficacy).

**Table 1 cancers-16-01190-t001:** Baseline characteristics.

		Valid Cases	Medical Treatment
Mean age (Y ± SD)		192	64 ± 13
Gender	Male	192	108 (56%)
Year of diagnosis		192	
	2000–2009		46 (24%)
	2010–2020		146 (76%)
Incidentaloma		185	59 (32%)
Functional tumor		192	
	Insulinoma		7 (4%)
	Gastrinoma		5 (3%)
	Other		6 (3%)
	Total functional tumors		18 (9%)
Ki-67 index, median		187	16 (8–50)
CgA, pmol/L, median		173	322 (98–1020)
Stage		192	
	Local		26 (14%)
	Regional		22 (11%)
	Metastatic		144 (75%)
WHO Grade		187	
	NET G1		17 (9%)
	NET G2		90 (48%)
	NET G3		20 (11%)
	NEC		60 (32%)
Location in pancreas		159	
	Caput		87 (56%)
	Cauda/corpus		72 (45%)
Surgery	Primary surgery	192	45 (23%)
	Palliative surgery	192	7 (4%)

Baseline characteristics of 192 patients receiving medical treatment. Y = years, SD = standard deviation, CgA = chromogranin A.

**Table 2 cancers-16-01190-t002:** HR and 95% CI for the Cox regressions.

SSA Progression-Free Survival			
	Univariable Analysis		
Variables	HR	95% CI	*p*-value
Age	1.0	1.0–1.0	0.979
Sex (ref. female)	1.1	0.5–2.2	0.877
WHO Grade (ref. NET G1)			0.101
NET G2	3.7	1.1–12.5	0.032
Log2(Ki-67)	1.6	1.1–2.3	0.025
Stage (ref. localized)			0.135
Regional	0.3	0.0–2.8	0.313
Disseminated	1.8	0.7–4.4	0.198
Size primary tumor	1.2	1.0–1.3	0.010
Line of treatment (ref. 1. line)	3.4	1.7–7.0	<0.001
	Multivariable analysis		
	(incl. Ki-67 index)		
Variables	HR	95% CI	*p*-value
Line of treatment (ref. 1. line)	4.1	1.9–9.1	<0.001
	Multivariable analysis		
	(incl.WHO Grade)		
Variables	HR	95% CI	*p*-value
Line of treatment (ref. 1. line)	3.1	1.4–6.9	0.006

Prognostic factors for progression-free survival in patients treated with somatostatin analogues (SSA). Table showing hazard ratio (HR) and 95% confidence interval (CI) for prognostic factors in both uni- and multivariant analyses. Multivariable analyses were performed with proliferation expressed as a categorical variable (WHO grade) or as a continuous variable (Ki-67 index).

**Table 3 cancers-16-01190-t003:** HR and 95% CI for the Cox regressions.

STZ/5FU Progression-Free Survival			
	Univariable Analysis		
Variables	HR	95% CI	*p*-value
Age	1.0	1.0–1.0	0.568
Sex (ref. female)	1.0	0.6–1.7	0.874
WHO Grade (ref. NET G1)			0.030
NET G2	2.8	1.1–7.2	0.031
NEC	11.1	2.1–60.9	0.005
Log2(Ki67)	1.3	1.1–1.6	0.008
Stage (ref. localized)			0.057
Regional	3.9	1.3–12.1	0.018
Disseminated	1.7	0.8–3.8	0.208
Primary operation	0.6	0.2–1.3	0.174
	Multivariable analysis		
	(incl. Ki-67 index)		
Variables	HR	95% CI	*p*-value
Log2(Ki67)	1.3	1.1–1.6	0.008
	Multivariable analysis		
	(incl.WHO Grade)		
Variables	HR	95% CI	*p*-value
WHO Grade (ref. NET G1)			0.030
NET G2	2.8	1.1–7.2	0.031
NEC	11.1	2.0–60.9	0.005

Prognostic factors for progression-free survival in patients treated with streptozocin/5-fluorouracil (STZ/5FU). Table showing hazard ratio (HR) and 95% confidence interval (CI) for prognostic factors in both uni- and multivariant analyses. Multivariable analyses were performed with proliferation expressed as a categorical variable (WHO grade) or as a continuous variable (Ki-67 index).

**Table 4 cancers-16-01190-t004:** HR and 95% CI for the Cox regressions.

PRRT Progression-Free Survival			
	Univariable Analysis		
Variables	HR	95% CI	*p*-value
Age	1.0	1.0–1.0	0.975
Sex (ref. female)	1.2	0.6–2.4	0.655
WHO Grade (ref. NET G1)			0.316
NET G2	2.3	0.8–6.6	0.135
NET G3	2.6	0.4–14.9	0.297
Log2(CgA)	1.2	1.0–1.4	0.055
Log2(Ki-67)	1.2	0.9–1.6	0.147
Stage (ref. localized)			0.250
Regional	1.4	0.3–6.2	0.672
Disseminated	2.3	0.8–6.8	0.123
	Multivariable analysis		
	(incl. Ki-67 index)		
Variables	HR	95% CI	*p*-value
Log2(CgA)	1.2	1.0–1.4	0.011
Log2(Ki-67)	1.4	1.0–1.8	0.052
	Multivariable analysis		
	(incl.WHO Grade)		
Variables	HR	95% CI	*p*-value
Log2(CgA)	1.2	1.0–1.4	0.055

Prognostic factors for progression-free survival in patients treated with peptide receptor radionuclide therapy (PRRT). Table showing hazard ratio (HR) and 95% confidence interval (CI) for prognostic factors in both uni- and multivariant analyses. Multivariable analyses were performed with proliferation expressed as a categorical variable (WHO grade) or as a continuous variable (Ki-67 index). CgA = chromogranin A.

**Table 5 cancers-16-01190-t005:** HR and 95% CI for the Cox regressions.

**(A) Temozolomide progression-free survival**			
	Univariable analysis		
Variables	HR	95% CI	*p*-value
Age	1.0	1.0–1.0	0.543
Sex (ref. female)	1.2	0.7–2.2	0.467
**(B) Carboplatin + etoposide progression-free survival**			
	Univariable analysis		
Variables	HR	95% CI	*p*-value
Age	1.0	1.0–1.0	0.740
Sex (ref. female)	1.1	0.7–1.2	0.620

Prognostic factors for progression-free survival in patients treated with temozolomide (A) or carboplatin and etoposide (B). Tables showing hazard ratio (HR) and 95% confidence interval (CI) for prognostic factors in univariant analyses.

## Data Availability

Data are contained within the article.

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
