# Peer review of "Responses to Medical Treatment in 192 Patients with Pancreatic Neuroendocrine Neoplasms Referred to the Copenhagen Neuroendocrine Tumour Centre in 2000–2020"

_cancers, 2024, doi:10.3390/cancers16061190_

Round 1

Reviewer 1 Report

Comments and Suggestions for Authors

The title is interesting and attracts reader's attention. Introduction is clear an offers sufficient information. Results presented and statistical data is abundant offering important insights into this subject. Discussion debates the main findings of this manuscript, integrating and comparing the obtained data with recent studies from literature and also with present guidelines, using up to date references. This article is hard worked and valuable and deserves to be published. However, moderate revisions are necessary to improve its quality and to offer a more readable and accessible variant of the manuscript. In this regard, even though the statical analysis is abundant, the authors should structure the data presented in a more clearly manner, so that the article is more easy to read.

In Introduction, lines 75-76, the authors state that in this study are presented data from 192 patients who received medical treatment for unresectable disease. The authors have to detail what they want to say  by unresectable. In lines 93-95 the stage is classified upon pathological findings - needs to be clarified if this refers to pathological findings after surgery or biopsy? - because if we discuss unresectable patients, surgery is not considered at all. Local disease and regional disease at baseline if included in unresectable have to be explained what were the "unresectable" criteria.

Lines 107-108 "In order to ensure meaningful statistical analyses, the evaluation of treatment responses  was limited to medical treatments that were administered to 20 or more patients within the cohort" Paragraph is not clear and authors have to clarify what they mean with "20 or more"

Lines 111-112 state that previous data on this cohort have been published analysing OS and median recurrence after intended radical surgery. I recommend the authors to separately analyse and present unresectable cases from unresectable recurrence of local disease after initial surgery or at least to clearly explain what they mean by unresectable.

Baseline characteristics of the cohort presented in table 1 should be explained in text.

Typographical errors should be revised. For example, lines 31 and 33 - "treat-ment" and "fac-tors" and line 35 "grade (G)1" . Minor English language editing is also necessary.

Comments on the Quality of English Language

Minor English language revisions required

Reviewer 2 Report

Comments and Suggestions for Authors

The article "Responses to medical treatment in 192 patients with pancreatic neuroendocrine neoplasms referred to the Copenhagen Neuroendocrine Tumour Centre in 2000-2020" presents a retrospective study on the efficacy of medical treatments for pancreatic neuroendocrine neoplasms (pNEN). It covers treatments and outcomes for 192 patients over two decades. The study found that somatostatin analogs were effective for low-grade tumors, streptozocin and 5-fluorouracil showed promising efficacy for G2 disease, and peptide receptor radionuclide therapy (PRRT) could be considered as a first-line treatment for NET G2. However, high-grade tumors had poor prognosis with carboplatin/etoposide or temozolomide. Median overall survival varied with tumor grade, and treatment line was an independent risk factor for time to progression.

Based on the article, the following shortcomings are identified:

 1. Retrospective Design

The study's retrospective nature is a fundamental limitation. Retrospective studies are inherently subject to biases, including selection bias and recall bias. These biases can affect the accuracy of the collected data and the generalizability of the study's findings. Prospective studies are more robust as they can be designed to minimize these biases from the outset.

 2. Single-Center Study

The research was conducted at a single center, the Copenhagen University Hospital - Rigshospitalet. While this allows for a consistent treatment approach and follow-up, it also limits the generalizability of the results. Treatment protocols, patient demographics, and healthcare systems vary globally, which might affect the applicability of the findings in different contexts.

 3. Limited Sample Size for Certain Treatments

The study mentions that the evaluation of treatment responses was limited to medical treatments administered to 20 or more patients within the cohort. This exclusion criterion omitted potentially valuable data on less commonly used treatments. Additionally, for treatments just meeting this threshold, the sample size might still need to be more significant to draw robust conclusions, especially for high-grade tumors where treatment options are limited and outcomes are generally poor.

 4. Lack of Randomization

As a retrospective study, there was no random assignment of treatments, which could lead to confounding variables influencing the outcomes. For example, patients with more aggressive diseases might be more likely to receive specific treatments, skewing the results. Randomized controlled trials are the gold standard for assessing treatment efficacy because they minimize the impact of confounding variables.

 5. Potential for Overfitting in Multivariable Analysis

The article mentions that Cox regression analyses were conducted exclusively for treatments administered to 40 or more patients to prevent overfitting of the statistical models. While this approach is prudent, it also suggests that the study might be at risk of overfitting when analyzing treatments close to this threshold. Overfitting can make the model too complex, capturing noise rather than the underlying relationship, which can affect the validity of the prognostic factors identified.

 6. Limited Exploration of Molecular and Genetic Factors

The study provides a comprehensive analysis of clinical and pathological prognostic factors but appears to have a limited focus on the molecular and genetic characteristics of the tumors. Given the increasing recognition of the role of genetic mutations and molecular markers in the behavior of neuroendocrine tumors and their response to treatment, a more detailed analysis in this area could provide valuable insights.

 7. Follow-up Duration

The median follow-up period was 28 months, which might not be sufficient to fully assess long-term outcomes, especially for low-grade tumors that tend to have a more indolent course. Longer follow-ups would provide more comprehensive data on survival and the long-term efficacy and safety of the treatments evaluated.

 8. Exclusion of Less Common Treatments

The study excluded medical treatments that were administered to 20 or fewer patients. This could lead to a need for more information on the efficacy and safety of these less common treatments, which might be relevant for specific patient subgroups or could become more widely used in the future.

 9. Limited Information on Interferon and Other Treatments

Specific treatments such as interferon alfa-2b, sunitinib, topotecan, and capecitabine were administered to a few patients (9, 8, 13, and 5, respectively) and were not included in the statistical analysis. This limits the study's ability to provide insights into these treatments' potential benefits or risks.

 10. Potential Bias in Treatment Line Analysis

The study found that treatment line was an independent risk factor for time to progression, with first-line treatments being used as a reference. However, the retrospective design may introduce bias, as the choice of treatment line could be influenced by factors not accounted for in the study, such as patient preferences, physician experience, or changes in treatment guidelines over time.

 11. Incomplete Data on Ki-67 Index

The Ki-67 index was unavailable in five patients, which could affect the accuracy of the study's findings related to tumor grade and prognosis, as Ki-67 is an essential marker for neuroendocrine tumor classification and grading.

 12. Variability in Disease Stage Classification

Based on pathological findings and imaging, the study classified the disease stage into three groups. However, this classification may not capture the full complexity of disease progression, and the study's retrospective nature could lead to inconsistencies in how the disease stage was determined at baseline.

 13. Reliance on Routine Clinical Practice for Progression Definition

Progression was defined based on death, clinical, or radiological progression assessed through routine clinical practice. This reliance on regular practice rather than standardized criteria could introduce variability in how progression is determined and reported.

 14. Use of Log Transformation for Continuous Variables

The study used log transformation for continuous variables like Ki-67 to achieve an approximate normal distribution. While this is a standard statistical technique, it can sometimes mask the true nature of the data distribution and may affect the interpretation of results.

 15. Lack of Data on Certain Patient Subgroups

The study may not provide detailed information on specific patient subgroups, such as those with rare functional tumors or specific genetic mutations, which could be necessary for personalized treatment approaches.

In summary, while the study provides valuable data on treating pancreatic neuroendocrine neoplasms, these limitations highlight the need for more comprehensive studies that include a wider range of treatments, standardized criteria for progression, and more detailed molecular and genetic analyses.

Comments on the Quality of English Language

minor
